# Exploring the interplay between BMI, subjective body image perception, and health behaviors: A cross-sectional study

**Ho-Jun Kim**[1☯], **In-Whi Hwang**[2☯], **Kyu-Ri Hong**[3], **Hae-Young Chung**[2], **Jung-Min Lee**[4,5*]

**1** Department of Physical Education, Graduate School of Physical Education, Kyung Hee University, Yongin-si, Republic of Korea, **2** Department of Sports Medicine and Science, Graduate School of Physical Education, Kyung Hee University, Yongin-si, Republic of Korea, **3** Department of Physical Education, Graduate School of Education, Kyung Hee University, Yongin-si, Republic of Korea, **4** Sports Science Research Center, Global Campus, Kyung Hee University, Yongin-si, Republic of Korea, **5** Department of Physical Education, Kyung Hee University, Yongin-si, Republic of Korea

☯ These authors contributed equally to this work.
* jungminlee@khu.ac.kr

## Abstract

### Background

This study explored the impact of discrepancies between Body Mass Index (BMI) and Subjective Body Image Perception (SBIP) on metabolic health indicators, physical activity (PA), sedentary behavior (SB), sleep time (ST), and stress levels in Korean adults.

### Methods

Data from 8,634 participants in the 8th Korea National Health and Nutrition Examination Survey (KNHANES, 2019–2021) were analyzed. Participants were categorized into three groups: Group A (SBIP=BMI), Group B (SBIP<BMI), and Group C (SBIP>BMI). Chi-square tests, ANOVA, and multinomial logistic regression were used to evaluate associations among discrepancies in SBIP and BMI and health behaviors.

### Results

Group B exhibited higher BMI levels (26.04 kg/m²) and adverse metabolic indicators, including elevated fasting glucose (102.11 mg/dL) and triglycerides (161.74 mg/dL), compared to the other groups ($p < 0.05$). Group C had better High-Density Lipoprotein (HDL) cholesterol (59 mg/dL) and lower prevalence rates of hyperlipidemia (9.7%) and hypertension (5.5%) than Group B (hyperlipidemia: 11.6%; hypertension: 5.1%) and Group A (hyperlipidemia: 13.2%; hypertension: 7.6%). Moderate-to-Vigorous PA (MVPA) was significantly lower in Group C (97.88 min/week) than Group A (133.18 min/week; $p < 0.05$) and Group B (169.64 min/week; $p < 0.05$). SBIP discrepancies

**Data availability statement:** All data files are available from the "https://knhanes.kdca.go.kr/knhanes/main.do".

**Funding:** This study was supported by the Ministry of Education of the Republic of Korea and the National Research Foundation of Korea (NRF) (NRF-2024S1A5A2A01023774 to J-ML). The funders had no role in study design, data collection and analysis, decision to publish, or preparation of the manuscript.

**Competing interests:** The authors have declared that no competing interests exist.

had a stronger effect on PA and SB than BMI alone, with Group C being 1.30 times more likely not to meet PA guidelines. Stress levels were significantly higher in those with lower BMI or higher SBIP (Odds Ratio [OR] = 1.93, $p < 0.01$).

## Conclusions

SBIP has a stronger influence on health behaviors, particularly PA patterns, than BMI alone. Including SBIP in health promotion strategies may improve interventions for improving PA and addressing metabolic health disparities.

---

## 1. Introduction

The Body Mass Index (BMI) is a widely recognized metric for classifying weight status and is closely associated with levels of physical activity (PA) [1]. Individuals with higher BMIs generally engage less in moderate-to-vigorous PA (MVPA), demonstrating an inverse relationship between BMI and PA participation [2]. However, BMI's scope is limited as it only accounts for height and weight [3], overlooking crucial psychological factors that influence health behaviors. One such factor is self-body image perception (SBIP), which encompasses an individual's subjective assessment of their body [4]. Negative body image perceptions (NBIP), such as dissatisfaction with one's appearance or concerns about weight, can contribute to chronic stress by causing emotional distress, lowering self-esteem [5], and triggering anxiety or depressive symptoms [6]. It may also increase social stress due to fear of judgment or rejection based on appearance [7].

Individuals with NBIP may face barriers to engaging in PA due to feelings of self-consciousness, embarrassment, or dissatisfaction with their physical appearance [8]. These psychological factors not only decrease motivation for PA, but also weaken belief in its benefits [9]. According to a study by Rech, NBIP and lack of confidence in PA accounted for 7.1% and 15.4% of general barriers to PA, respectively [10].

NBIP can also negatively impact metabolic health indicators such as elevated fasting glucose (FG), Hemoglobin A1c (HbA1c), cholesterol levels, and blood pressure [11,12], often mediated through reduced PA participation [13]. In contrast, a positive body image perception is often associated with better adherence to healthy behaviors, including regular participation in PA and balanced nutrition leading to improved metabolic profiles [14,15].

Globally technology advancement and changes in work patterns have reduced PA and increased sedentary behavior (SB), contributing to the rise in obesity rates worldwide [16]. Extended SB — such as long hours spent driving — has been linked to a significantly higher risk of cardiovascular mortality [17]. In 2016, approximately 26.5% of the adults failed to meet recommended PA levels, and obesity rates have more than doubled since 1990 [18,19], with future increases anticipated [20].

Psychological mechanisms underlying SBIP, such as societal standards of beauty, self-esteem, and internalized social comparisons, play a pivotal role in shaping health behaviors [21,22]. These factors can motivate healthier lifestyle choices or lead to

detrimental behaviors like overeating, smoking, and excessive alcohol consumption, which adversely affect markers of metabolic health [23–25]. Furthermore, the association between BMI and PA participation is compounded by psychological factors such as SBIP. Recent studies have highlighted that SBIP significantly influences PA engagement, with findings showing that a substantial percentage of individuals report SBIP as a determinant of their PA habits [26,27]. However, inconsistencies between BMI and SBIP can have substantial impacts on both PA participation and mental health. This demonstrates the need to consider both physical and psychological factors in health assessments.

Socioeconomic status (SES) and cultural background influence SBIP and access to health-promoting resources. Lower SES is often associated with a higher rates of negative body image and poorer health outcomes due to limited access to exercise facilities, and healthcare services [28–30]. Additionally, inadequate sleep time (ST), defined as fewer than seven hours per night, has been linked to higher likelihood of coronary heart disease, diabetes, and other severe health conditions, in contrast to the protective effects conferred by 7–8 hours of sleep [31]. Given the profound associations observed, the maintenance of equilibrium among PA, SB, and ST stands as imperative for fostering both mental and physical health.

This study investigates how the degree of inconsistency between individuals' BMI and SBIP influences metabolic health indicators, PA, SB, ST, and stress levels. While prior research has examined the impact of BMI and SBIP on health behaviors independently, this study uniquely integrated physical and psychological health factors to comprehensively understand of how these elements influence daily health behaviors and metabolic health [13,32,33].

## 2. Methods

### 2.1. Study participants

The study utilized raw data from the 8th Korea National Health and Nutrition Examination Survey (KNHANES 8th), conducted between 2019 and 2021. The KNHANES is a cross-sectional survey designed to evaluate the health and nutritional status of the Korean population [34]. The dataset from the 8th KNHANES included responses from 22,559 individuals. Fig 1 illustrates the flow diagram of participant selection. For the purposes of this study, we focused on working adults and thus, excluded 3,868 participants who were under 20 years old and 7,177 who were 60 years old or older. Additionally, we excluded 1,184 individuals who responded with 'don't know' or did not respond to critical variables. Since the study aimed to investigate individuals with normal, overweight, and obese BMI or SBIP categories, we further excluded 1,696 people with underweight BMI or an SBIP of thin or underweight. After applying these exclusion criteria, a total of 8,634 participants remained in the study.

### 2.2. Ethical approval

This study utilized data from the 8th KNHANES (2019–2021). The survey protocols were approved by the Institutional Review Board (IRB) of the Korean Centers for Disease Control and Prevention (KCDC) for each year: 2019 (IRB No. 2018-01-03-C-A), 2020 (IRB No. 2018-01-03-2C-A), and 2021 (IRB No. 2018-01-03-5C-A). The study adhered to the ethical principles outlined in the Declaration of Helsinki and its subsequent amendments. The KNHANES data are fully anonymized and publicly available. Written informed consent was obtained from all participants by the Korea Centers for Disease Control and Prevention (KCDC) at the time of data collection. For this secondary analysis, the need for additional ethical approval and consent was waived, as the data do not contain any identifiable personal information. The funders had no role in study design, data collection and analysis, decision to publish, or preparation of the manuscript.

### 2.3. Measures

#### 2.3.1. Anthropometrics measurements. KNHANES gathers data via self-reported surveys and direct physical measurement using standardized instrumentation. Age and gender were obtained through self-reported questionnaires, while anthropometric data were collected by trained medical staff during mobile health examination sessions. All physical

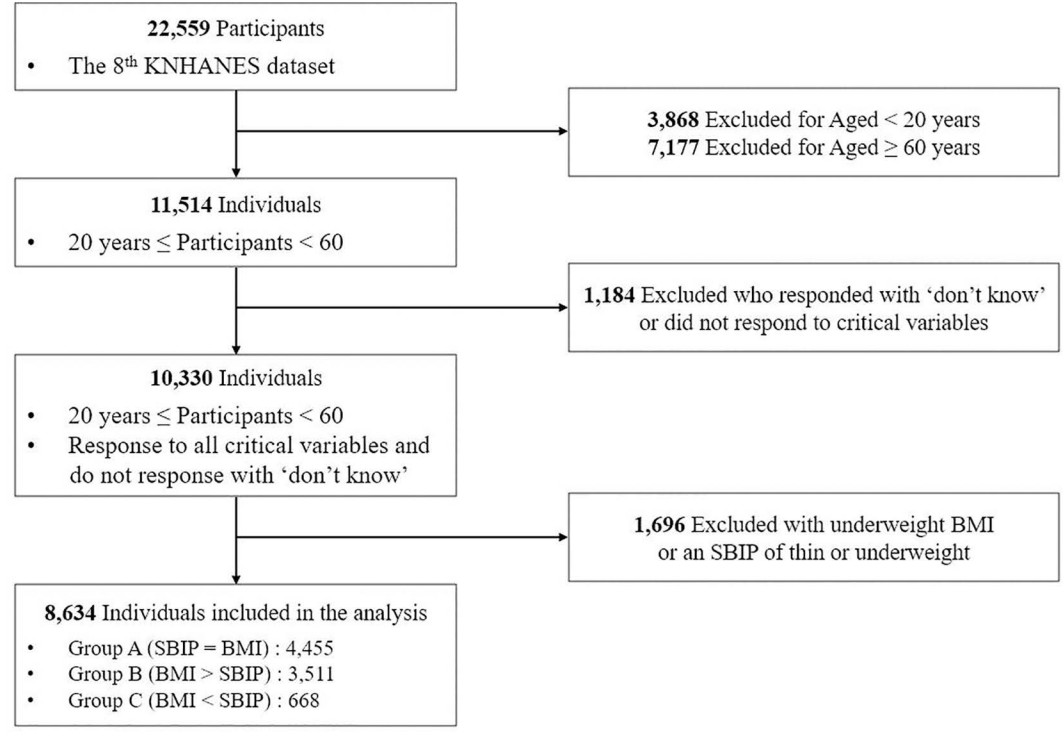

**Fig 1. Flow diagram.**

measurements were taken twice, with a brief interval between trials, and recorded by a designated recorder to ensure consistency and accuracy.

Height was measured in centimeters using a SECA 274 stadiometer (SECA, Germany) with participants standing upright in the Frankfurt plane position. Weight was measured in kilograms to the nearest 0.1 kg using a GL-6000-20 scale (G-tech, South Korea), with participants wearing light clothing and no shoes. Waist circumference (WC) was measured in centimeters at the midpoint between the lower margin of the rib cage and the iliac crest, after a normal exhalation, using W606PM tape measure (Lufkin, USA). BMI was calculated by dividing the weight (kg) by the square of the height (m²).

**2.3.2. Blood pressure measurements.** Blood pressures were measured using the WatchBP Office AFIB device (Microlife, Switzerland). Measurements included both systolic and diastolic blood pressure, and each participants' blood pressure was recorded in triplicate. For analysis, the average of the second and third readings of systolic and diastolic blood pressure was used. The values were recorded to the nearest whole number in mmHg. During the analysis phase, the mean and standard deviation for the systolic and diastolic blood pressures were calculated and presented to two decimal places.

**2.3.3. Socioeconomic status.** The socioeconomic factors considered in this study included; Type of residence (Single-family house, Apartments, Others), Educational Level (Under Elementary school graduate, Middle-school graduate, High-school graduate, Over university graduation), Occupational status, and Income quintile (low, lower middle, middle, upper middle, high).

Occupational status was classified into four categories based on the characteristics and typical work environments of the participants' jobs. White-collar workers were identified as salaried professionals and office workers, such as managers and clerks, whose roles involve administrative or professional tasks not directly tied to physical production, while

blue-collar workers were those engaged in manual labor, such as factory workers and laborers, typically in industrial or construction settings where they contribute directly to the production of goods. Pink-collar workers were defined as individuals in service-oriented or caregiving roles traditionally associated with women, such as nurses, teachers, and retail service workers, often requiring interpersonal and emotional skills. Occupations that could not be categorized due to insufficient or ambiguous information in the survey responses were labeled as not classified.

**2.3.4. Lifestyle behaviors.** The lifestyle behaviors assessed in this study included drinking and smoking status. Drinking status was categorized into four groups (Abstained in the last year, Once a month, 2–4 times a month, At least twice a week) [35] and smoking status was categorized into three groups (Never smoked, Ex-smoker:, Current smoker). Ex-smokers were defined as individuals who had smoked in the past but were not smoking at the time of the survey.

**2.3.5. Group.** The categorization of obesity levels based on BMI adhered to the standards set by Korean Society for the Study of Obesity BMI classification: Normal (18.5 kg/m$^2$ ≤ BMI < 23 kg/m$^2$), Overweight (23 kg/m$^2$ ≤ BMI < 25 kg/m$^2$), Obese (25 kg/m$^2$ < BMI). SBIP was classified based on participants' responses (very underweight, slightly underweight, normal, slightly obese, very obese). Participants who reported being 'slightly obese' were categorized as overweight, while those reporting as 'very obese' were classified as obese. This classification was based on actual BMI measurements in conjunction with participants' SBIP. Participants were then divided into three groups based on the congruence between their BMI and SBIP categories: (1) Group A: individuals whose BMI and SBIP classifications were concordant, falling into the same category of normal, overweight, or obese, (2) Group B: individuals classified as obese by BMI but overweight or normal by SBIP, and those classified as overweight by BMI but normal by SBIP. (3) Group C: individuals classified as obese by SBIP while being overweight or normal by BMI, and those classified as overweight by SBIP but normal by BMI. Body image misperception and related behavioral outcomes, such as changes in PA, are commonly observed among individuals with normal, overweight, obese [36]. In contrast, underweight individuals tend to exhibit consistently lower levels of PA and other distinct characteristics that differentiate them from these groups [37,38]. Therefore, participants with a BMI below 18.5 kg/m$^2$ and those who underweight or thin in their SBIP were excluded from this study.

**2.3.6. Health-related factors.** MVPA and SB were assessed using the Korean Version of Global Physical Activity Questionnaire, validated for Korean populations [39]. ST was measured using specific questions from the KNHANES.

**(1) Moderate to vigorous physical activity (MVPA).**

In this study, MVPA was calculated by summing four distinct types of physical activity: work-related high-intensity, work-related moderate-intensity, leisure-time high-intensity, and leisure-time moderate-intensity. Participants were asked to report their weekly engagement in each activity type, and the total minutes of MVPA per week were calculated using the equation:

$$\text{Total minutes per week} = (\text{Hours per day} \times 60 + \text{Additional minutes per day}) \times \text{Number of days}$$

To account for intensity differences, vigorous physical activity (VPA) minutes were weighted as twice the value of moderate physical activity (MPA) minutes before summing the total MVPA.

**(2) Sedentary behavior (SB)**

SB was calculated based on responses to two questions: (1) How many hours do you spend sitting on a typical day? (2) How many additional minutes do you spend sitting on a typical day?

To convert the total sedentary time into weekly minutes, the following calculation was used:

$$\text{Total weekly sedentary minutes} = (\text{Hours per day} \times 60 + \text{Additional minutes per day}) \times 7$$

### (3) Sleep time (ST)

ST was assessed using responses to two specific questions: (1) What is your average sleep duration per day during weekdays (or workdays)? (2) What is your average sleep duration per day during weekends (or non-workdays, including the day before a non-workday)?

To calculate the average weekly sleep duration, responses to the weekday sleep question were multiplied by 5, and responses to the weekend sleep question were multiplied by 2.

The total weekly sleep time was then converted into minutes by multiplying by 60, as shown in the following equation:

$$Total\ weekly\ sleep\ minutes = [(Weekly\ sleep\ duration \times 5) + (Weekend\ sleep\ duration \times 2)] \times 60$$

In 2021, the accuracy and reliability of sleep duration estimates were increased by subdividing the questions as follows: (1) Time of going to bed on weekdays (hours), (2) Time of going to bed on weekdays (minutes), (3) Time of waking up on weekdays (hours), (4) Time of waking up on weekdays (minutes), (5) Time of going to bed on weekends (hours), (6) Time of going to bed on weekends(minutes), (7) Time of waking up on weekends (hours), (8) Time of waking up on weekends (minutes).

Minutes were converted into hours by dividing by 60 to harmonize the measurements and facilitate calculations.

### (4) Perceive stress level

Stress level was assessed using the question: How much stress do you feel daily?

Participants were divided into four groups based on their responses (Rarely, Occasionally, Often, Very Often).

### (6) Metabolic health indicators

Participants were asked to fast overnight prior to the blood draw to ensure the accuracy of the test results. The blood samples were analyzed for FG, HbA1c, total cholesterol, HDL cholesterol, hypertension, and triglycerides. The classification of these parameters followed the criteria by the World Health Organization (WHO). FG: normal (70–99 mg/dL), impaired FG (100–125 mg/dL), diabetes (≥ 126 mg/dL); HbA1c: normal (< 5.7%), borderline (5.7–6.4%), diabetes (≥ 6.5%); Total cholesterol: normal (< 200 mg/dL), borderline (200–239 mg/dL), hyperlipidemia (≥ 240 mg/dL); HDL cholesterol: low (< 40 mg/dL), normal (40–59 mg/dL), high (≥ 60 mg/dL); Hypertension: normal (systolic < 120 mmHg and diastolic < 80 mmHg), prehypertension (systolic 120–139 mmHg or diastolic 80–89 mmHg), hypertension (systolic ≥ 140 mmHg or diastolic ≥ 90 mmHg); Triglycerides: normal (< 150 mg/dL), borderline (150–199 mg/dL), high (≥ 200 mg/dL).

**2.3.7. Data analysis.** All data processing and statistical analyses were performed using the SPSS 28.0 version (SPSS Inc., Chicago, IL, USA). Descriptive statistics were employed to summarize participants' age, gender, anthropometric variables (height, weight, and BMI), systolic blood pressure, diastolic blood pressure, FG, HbA1c, total cholesterol, HDL cholesterol, hypertension, and triglycerides. Chi-square ($\chi^2$) analysis was used to compare categorical variables of general characteristics among the three groups. Prior to performing the Chi-square tests, we ensured that all expected cell counts met the minimum requirement of 5, thus satisfying the assumptions required for the chi-square analysis. For the evaluation of differences in time spent in MVPA, SB, and ST across the groups, ANOVA was conducted. The Bonferroni post-hoc test was subsequently applied for pairwise comparisons to determine specific group differences. To examine the factors associated with discrepancies between SBIP and BMI, multinomial logistic regression analysis was performed. The dependent variable was the SBIP and BMI discrepancy group, categorized as Group A, Group B, and Group C. Group A was set as the reference category in the analysis. Independent variables included educational level, occupational status, individual income quintile, drinking and smoking status, perceived stress level, moderate-to-vigorous physical activity (MVPA), sleep duration, sedentary behavior, and health indicators such as FG, HbA1c, total cholesterol, HDL cholesterol, hypertension, and triglyceride levels. Independent variables were selected based on theoretical relevance and

their associations in univariate analyses ($p<0.20$). Multicollinearity was assessed using variance inflation factors (VIF), and all variables showed VIF values below 1.6, indicating no multicollinearity concerns. The significance level was set at $p=0.05$ for calculating the multinomial logistic regression, and the results were expressed in terms of odds ratios (OR) with 95% confidence intervals (CI). No additional covariate adjustments were made in the models.

## 3. Results

Table 1 presents the characteristics and anthropometric measurements for participants divided into three groups (Group A: $n=4,455$, Group B: $n=3,511$, Group C: $n=668$) using descriptive statistics. The results are summarized in number and proportion in Table 1. Overall, the gender distribution varied significantly. Group A and Group C had more females (70.1% and 92.2%, respectively), while Group B had more males (69.7%). In Group A and Group B, the proportion of the group increased with age, but in Group C, the proportion of 40–49 years was the highest. Anthropometric measurements revealed differences between the groups. The mean age was 41.6 years in Group A, 43.19 years in Group B, and 39.07 years in Group C. Group B participants were the tallest (169.02 cm), followed by Group A (164.40 cm) and Group C (162.13 cm). Weight was highest in Group B (74.60 kg) compared to Group A (65.56 kg) and Group C (57.95 kg). Similarly, WC and BMI were highest in Group B. Systolic blood pressure showed that Group B had the highest mean (119.23 mmHg) followed by Group A (114.06 mmHg) and Group C (109 mmHg). In diastolic blood pressure, Group B (78.79 mmHg) was followed by Group A (75.17 mmHg) and Group C (71.66 mmHg). FG levels were highest in Group B (102.11 mg/dL) and lowest in Group C (93.76 mg/dL). Total cholesterol levels were similar across the groups, but HDL cholesterol was higher in Group C (59 mg/dL) compared to Group A (54.83 mg/dL) and Group B (49.2 mg/dL). Triglyceride levels

**Table 1. Characteristics and anthropometrics for participants in three groups ($n=8,634$).**

| Variables | | Group A ($n=4,455$) | | Group B ($n=3,511$) | | Group C ($n=668$) | |
|---|---|---|---|---|---|---|---|
| | | NO. (%) | Mean ± SD | NO. (%) | Mean ± SD | NO. (%) | Mean ± SD |
| **Gender** | Male | 1,330 (29.9) | | 2,448 (69.7) | | 52 (7.8) | |
| | Female | 3,125 (70.1) | | 1,063 (30.3) | | 616 (92.2) | |
| **Age (year)** | 20–29 | 836 (18.8) | 24.51 ± 2.79 | 557 (15.9) | 24.97 ± 2.65 | 171 (25.6) | 24.61 ± 2.82 |
| | 30–39 | 1,004 (22.5) | 35.05 ± 2.94 | 705 (20.1) | 35.18 ± 2.85 | 158 (23.7) | 34.73 ± 2.94 |
| | 40–49 | 1,295 (29.1) | 44.46 ± 2.87 | 1,030 (29.3) | 44.78 ± 2.92 | 204 (30.5) | 44.44 ± 2.96 |
| | 50–59 | 1,320 (29.6) | 54.61 ± 2.88 | 1,219 (34.7) | 54.8 ± 2.84 | 135 (20.2) | 54.36 ± 2.86 |
| **Anthropometrics** | Age (year)*,† | 4,455 | 41.6 ± 11.19 | 3,511 | 43.19 ± 11.01 | 668 | 39.07 ± 11.07 |
| | Height (cm)*,† | | 164.4 ± 8.32 | | 169.02 ± 8.98 | | 162.13 ± 6.53 |
| | Weight (kg)*,† | | 65.56 ± 15.10 | | 74.6 ± 10.08 | | 57.95 ± 5.43 |
| | Waist circumference (cm)*,† | | 82.26 ± 11.96 | | 88.86 ± 6.83 | | 76.36 ± 5.23 |
| | BMI (kg/m$^2$)*,† | | 24.10 ± 4.40 | | 26.04 ± 2.13 | | 21.98 ± 1.08 |
| | Systolic Blood Pressure (mmHg)*,† | | 114.06 ± 14.33 | | 119.23 ± 13.56 | | 109 ± 13.46 |
| | Diastolic Blood Pressure (mmHg)*,† | | 75.17 ± 10.00 | | 78.79 ± 9.71 | | 71.66 ± 9.48 |
| | Fasting Glucose (mg/dL)*,† | | 98.16 ± 21.07 | | 102.11 ± 23.01 | | 93.76 ± 16.13 |
| | Total Cholesterol (mg/dL)* | | 196.3 ± 36.94 | | 198.42 ± 37.51 | | 195.82 ± 35.58 |
| | HDL Cholesterol (mg/dL)*,† | | 54.83 ± 13.24 | | 49.2 ± 11.47 | | 59 ± 12.48 |
| | Triglyceride (mmHg)*,† | | 121 ± 96.93 | | 161.74 ± 143.94 | | 98.3 ± 67.01 |
| | HbA1c (%)*,† | | 5.68 ± 0.85 | | 5.82 ± 0.88 | | 5.47 ± 0.64 |

*Note.* SD: Standard deviation, BMI: Body Mass Index,

* significant difference ($p<0.05$) between Group A and Group B,

† significant difference ($p<0.05$) between Group B and Group C.

Group A: SBIP = BMI, Group B: BMI > SBIP, Group C: BMI < SBIP.

were highest in Group B (161.74 mg/dL) and lowest in Group C (98.3 mg/dL). HbA1c levels were slightly higher in Group B (5.82%) compared to Groups A (5.68%) and Group C (5.47%). Among the anthropometric measurements, there were significant differences between Group A and Group B, and between Group B and Group C in all variables except total cholesterol, which showed a significant difference only between Group A and Group B.

Table 2 shows the distribution of demographic attributes, health indicators, and PA across three distinct groups. The analysis identified a significant variation in educational attainment ($p < 0.001$), with a notably low prevalence of individuals possessing education levels below elementary school and middle school graduation across all groups. The occupational status also exhibited significant differences among the three groups ($p < 0.001$), where Group A and Group B had a higher proportion of blue-collar workers, whereas Group C was predominantly composed of non-classified workers. In contrast, no significant differences were observed in the individual income quintile. Drinking ($p < 0.001$) and smoking ($p < 0.001$) statuses differed significantly across the groups. Perceived stress levels also exhibited significant discrepancies ($p < 0.001$), with occasional stress being the most prevalent across all groups, followed by frequent stress. Health-related variables, such as FG ($p < 0.001$), HbA1c ($p < 0.001$), total cholesterol ($p = 0.003$), HDL cholesterol ($p < 0.001$), hypertension ($p < 0.001$), triglyceride ($p < 0.001$), and systolic blood pressure ($p < 0.001$), also differed significantly across the three groups. Compliance with PA guidelines showed significant variance ($p < 0.01$). Additionally, ST varied significantly across the groups ($p < 0.001$), with the majority reporting 7–9 hours of sleep in all groups. SB ($p < 0.001$), and the frequency of walking days per week ($p = 0.001$) also exhibited significant differences among the groups.

Table 3 shows the result of multinomial logistic regression analysis, assessing the association between various demographic and health-related variables, PA, and SB, and the differences in SBIP and BMI among different groups. The analysis revealed significant findings regarding occupational status and lifestyle behaviors. Using Group A as the reference category, comparted to Group A, participants in Group B were more likely to be white-collar workers (OR = 1.26; 95% CI = 1.08–1.47; $p < 0.01$) and blue-collar workers (OR = 1.46; 95% CI = 1.29–1.64; $p < 0.001$). Interestingly, Groups A and C showed a lower likelihood of blue-collar occupation (OR = 0.64; 95% CI = 0.47–0.85; $p < 0.001$). When it came to drinking habits, individuals in Groups A and B were more likely to consume alcohol at least twice a week (OR = 1.21; 95% CI = 1.04–1.42; $p < 0.05$). Smoking status also varied significantly: ex-smokers were more chance of disease in Group B compared to group A (OR = 2.16; 95% CI = 1.91–2.44; $p < 0.001$), while Group C had fewer ex-smokers than Group A (OR = 0.75; 95% CI = 0.58–0.96; $p < 0.05$). Current smokers followed a similar trend with higher odds in Groups A and B (OR = 2.07; 95% CI = 1.82–2.36; $p < 0.001$) and lower odds in Group A and C (OR = 0.64; 95% CI = 0.47–0.85; $p < 0.01$). Stress levels varied among the groups. Group B individuals were less likely to experience occasional stress (OR = 0.80; 95% CI = 0.65–0.99; $p < 0.05$), frequent stress (OR = 0.67; 95% CI = 0.56–0.80; $p < 0.001$), and very frequent stress OR = 0.50; 95% CI = 0.38–0.64; $p < 0.001$) compared to Group A. Conversely, Group C had a higher likelihood of experiencing very frequent stress (OR = 1.93; 95% CI = 1.29–2.87; $p < 0.01$). Health indicators also showed significant associations. Group A and Group B were more likely to have borderline FG levels (OR = 1.17; 95% CI = 1.04–1.31; $p < 0.01$) and borderline HbA1c levels (OR = 1.15; 95% CI = 1.04–1.28; $p < 0.05$). They also showed a higher chance of low HDL cholesterol (OR = 1.30; 95% CI = 1.12–1.50; $p < 0.001$) and a lower chance of high HDL cholesterol (OR = 0.58; 95% CI = 0.51–0.65; $p < 0.001$). Similarly, they were more likely to have prehypertension (OR = 1.50; 95% CI = 1.35–1.66; $p < 0.001$) and hypertension (OR = 1.40; CI = 1.20–1.63; $p < 0.001$), as well as borderline (OR = 1.52; 95% CI = 1.32–1.76; $p < 0.001$) and high triglyceride levels (OR = 1.32; 95% CI = 1.14–1.53; $p < 0.001$). On the other hand, Group A and Group C exhibited different trends: they were less likely to have borderline FG (OR = 0.64; 95% CI = 0.50–0.82; $p < 0.001$), low HDL cholesterol (OR = 0.49; 95% CI = 0.32–0.75; $p < 0.001$), and prehypertension (OR = 0.60; 95% CI = 0.48–0.74; $p < 0.001$). but more likely to have high HDL cholesterol (OR = 1.44; 95% CI = 1.20–1.72; $p < 0.001$). In terms of PA, Group A and B were less likely to meet PA guidelines (OR = 0.77; 95% CI = 0.69–0.86; $p < 0.001$) and more likely to have insufficient ST (less than 7 hours) (OR = 1.12; 95% CI = 1.01–1.23; $p < 0.05$). In contrast, Group C was more likely to not meet PA guidelines (OR = 1.30; 95% CI = 1.06–1.58; $p < 0.05$) compared to Group A.

**Table 2. Result of chi-square analysis examining differences in general characteristics across three groups (n = 8,634).**

| Variables | | Group A (%) | Group B (%) | Group C (%) | *p*-value |
|---|---|---|---|---|---|
| **Educational Level** | Under Elementary school graduate | 76 (1.7) | 88 (2.5) | 2 (0.3) | < 0.001 |
| | Middle-school graduate | 190 (4.3) | 204 (5.8) | 22 (3.3) | |
| | High-school graduate | 1,740 (39.1) | 1,419 (40.4) | 250 (37.4) | |
| | Over University graduation | 2,449 (55.0) | 1,800 (51.3) | 394 (59.0) | |
| **Occupational Status** | White | 748 (16.8) | 598 (17.0) | 137 (20.5) | < 0.001 |
| | Blue | 1,845 (41.4) | 1,840 (52.4) | 223 (33.4) | |
| | Pink | 426 (9.6) | 304 (8.7) | 67 (10.0) | |
| | Not classified | 1,436 (32.2) | 769 (21.9) | 241 (36.1) | |
| **Income Quintile (Individual)** | Middle | 906 (20.3) | 718 (20.5) | 131 (19.6) | 0.271 |
| | Low | 841 (18.9) | 666 (19.0) | 126 (18.9) | |
| | Lower middle | 854 (19.2) | 745 (21.2) | 119 (17.8) | |
| | Upper middle | 937 (21.0) | 691 (19.7) | 148 (22.2) | |
| | High | 917 (20.6) | 691 (19.7) | 144 (21.6) | |
| **Drinking Status** | Abstained in the last year | 907 (20.4) | 597 (17.0) | 128 (19.2) | < 0.001 |
| | Once a month | 1,558 (35.0) | 967 (27.5) | 225 (33.7) | |
| | 2–4 times a month | 1,143 (25.7) | 956 (27.2) | 188 (28.1) | |
| | At least twice a week | 847 (19.0) | 991 (28.2) | 127 (19.0) | |
| **Smoking Status** | Never smoked | 3,037 (68.2) | 1,528 (43.5) | 516 (77.2) | < 0.001 |
| | Ex-smoker | 746 (16.7) | 1,049 (29.9) | 87 (13.0) | |
| | Current smoker | 672 (15.1) | 934 (26.6) | 65 (9.7) | |
| **Perceived Stress Level** | Almost Never | 443 (9.9) | 407 (11.6) | 63 (9.4) | < 0.001 |
| | Occasionally | 2,561 (57.5) | 2,096 (59.7) | 340 (50.9) | |
| | Often | 1,189 (26.7) | 850 (24.2) | 205 (30.7) | |
| | Very Often | 262 (5.9) | 158 (4.5) | 60 (9.0) | |
| **Fasting Glucose (md/dL)** | Normal | 3,149 (70.7) | 2,062 (58.7) | 562 (84.1) | < 0.001 |
| | Borderline | 1,063 (23.9) | 1,173 (33.4) | 94 (14.1) | |
| | Diabetes | 243 (5.5) | 276 (7.9) | 12 (1.8) | |
| **HbA1c (%)** | Normal | 1,957 (43.9) | 1,197 (34.1) | 379 (56.7) | < 0.001 |
| | Borderline | 2,219 (49.8) | 2,014 (57.4) | 275 (41.2) | |
| | Diabetes | 279 (6.3) | 300 (8.5) | 14 (2.1) | |
| **Total Cholesterol (md/dL)** | Normal | 2,506 (56.3) | 1,849 (52.7) | 387 (57.9) | 0.003 |
| | Borderline | 1,430 (32.1) | 1,198 (34.1) | 216 (32.3) | |
| | Hyperlipidemia | 519 (11.6) | 464 (13.2) | 65 (9.7) | |
| **HDL Cholesterol (md/dL)** | Normal | 2,506 (56.3) | 2,213 (63) | 330 (49.4) | < 0.001 |
| | Low | 497 (11.2) | 691 (19.7) | 27 (4) | |
| | High | 1,452 (32.6) | 607 (17.3) | 311 (46.6) | |
| **Hypertension** | Normal | 2,713 (60.9) | 1,555 (44.3) | 509 (76.2) | < 0.001 |
| | Borderline | 1,297 (29.1) | 1,421 (40.5) | 122 (18.3) | |
| | High | 445 (10) | 535 (15.2) | 37 (5.5) | |
| **Triglyceride (mmHg)** | Normal | 3,420 (76.8) | 2,098 (59.8) | 564 (84.4) | < 0.001 |
| | Borderline | 485 (10.9) | 638 (18.2) | 62 (9.3) | |
| | Danger | 550 (12.3) | 775 (22.1) | 42 (6.3) | |
| **Systolic Blood Pressure (mmHg)** | Normal | 3,064 (68.8) | 1,943 (55.3) | 551 (82.5) | < 0.001 |
| | Prehypertension | 1,163 (26.1) | 1,301 (37.1) | 95 (14.2) | |
| | Hypertension | 228 (5.1) | 267 (7.6) | 22 (3.3) | |

*(Continued)*

**Table 2.** (Continued)

| Variables | | Group A (%) | Group B (%) | Group C (%) | p-value |
|---|---|---|---|---|---|
| **PA Guideline Compliance** | Met | 1,127 (25.3) | 1,030 (29.3) | 149 (22.3) | < 0.001 |
| | Not met | 3,328 (74.7) | 2,481 (70.7) | 519 (77.7) | |
| **Sleep Time (hours/day)** | 7–9 hours | 2,437 (54.7) | 1,830 (52.1) | 362 (54.2) | < 0.001 |
| | Under 7 hours | 1,755 (39.4) | 1,523 (43.4) | 248 (37.1) | |
| | Over 7 hours | 263 (5.9) | 158 (4.5) | 58 (8.7) | |
| **Sedentary Behavior** | Lower | 1,410 (31.6) | 1,174 (33.4) | 180 (26.9) | < 0.001 |
| | Middle | 997 (22.4) | 848 (24.2) | 166 (24.9) | |
| | Higher | 2,048 (46) | 1,489 (42.4) | 322 (48.2) | |
| **Walking Days (wk)** | Not participating | 636 (14.3) | 568 (16.2) | 92 (13.8) | 0.001 |
| | 1–3 days | 1,370 (30.8) | 1,018 (29.0) | 245 (36.7) | . |
| | 4–6 days | 1,241 (27.9) | 940 (26.8) | 171 (25.6) | |
| | Everyday | 1,208 (27.1) | 985 (28.1) | 160 (24.0) | |

**Note.** The values are presented as n (%). *** $p < 0.001$, ** $p < 0.01$. Group A: SBIP = BMI, Group B: BMI > SBIP, Group C: BMI < SBIP, PA: Physical Activity.

In Fig 2, the results of one-way ANOVA revealed significant differences in the mean volume of MVPA per week across the three groups. Group B demonstrated a notably higher mean MVPA (169.64 ± 354.30 min/week) compared to Group A (133.18 ± 337.98 min/week) and Group C (97.88 ± 213.67 min/week) with the differences being statistically significant ($p < 0.001$). Additionally, a significant difference was also found between Group A and Group C ($p < 0.05$). Fig 3 presents a comparative analysis of the average SB among the groups. Group B had an average SB of 3564.87 ± 1490.43 min/week, which was higher than that of Group A (3645.75 ± 1497.38 min/week) and Group C (3787.60 ± 1487.26 min/week). Significant differences in SB were found between Group B and Group A ($p < 0.05$) and between Group B and Group C ($p < 0.01$). However, the difference between Group A and Group C was not statistically significant ($p = 0.07$). Fig 4 illustrates the comparison of average ST between the three groups. Group C reported the highest average ST (3024.78 ± 529.65 min/week), followed by Group A (2961.71 ± 337.98 min/week) and Group B (2914.49 ± 496.96 min/week). Statistically significant differences were observed between all group comparisons: Group A vs. Group B ($p < 0.001$), Group A vs. Group C ($p < 0.001$), and Group B vs. Group C ($p < 0.001$).

## 4. Discussion

The primary aim of this study was to examine the impact of the discrepancy between actual BMI and SBIP on various health indicators, including metabolic health indicators, PA, SB, ST, and stress levels. While previous research has primarily focused on BMI or SBIP as independent variables, this study addresses a critical gap by investigating how the difference between actual BMI and SBIP influences health outcomes. Prior studies have consistently shown that individuals with a higher BMI often exhibit decreased PA, prolonged SB, and elevated metabolic abnormalities, such as insulin resistance, dyslipidemia, and hypertension. However, the role of the discrepancy between actual BMI and SBIP in further influencing these health indicators remains underexplored. This study aims to provide a more comprehensive understanding of the interplay between BMI, SBIP, and various health outcomes, highlighting the potential for this discrepancy to serve as a significant determinant of health behaviors and metabolic vulnerability.

Our findings indicate that the chance of unfavourable metabolic markers decreases with increased PA and reduced SB. Specifically, for FG and HbA1c levels, participants with lower SBIP than BMI were more likely to be classified as obese. Conversely, those who perceived themselves as heavier than their actual BMI exhibited higher HDL cholesterol levels compared to participants who viewed themselves as thinner or the same as their actual BMI. Additionally,

**Table 3. Multinomial logistic regression results (*n* = 8,634).**

| Variables | | Group B | | | | Group C | | | |
|---|---|---|---|---|---|---|---|---|---|
| | | B | S.E | Wald | O.R (95% CI) | B | S.E | Wald | O.R (95% CI) |
| **Educational Level** | Under Elementary school graduate | Ref. | | | | Ref. | | | |
| | Middle-school graduated | −0.10 | 0.20 | 0.24 | 0.91 [0.62–1.34] | 1.42 | 0.76 | 3.52 | 4.12 [0.94–18.11] |
| | High-school graduate | −0.26 | 0.17 | 2.27 | 0.77 [0.55–1.08] | 1.41 | 0.72 | 3.79 | 4.10 [0.99–16.95] |
| | Over University graduate | −0.32 | 0.17 | 3.42 | 0.73 [0.52–1.02] | 1.41 | 0.72 | 3.80 | 4.11 [0.99–16.99] |
| **Occupational Status** | Not classified | Ref. | | | | Ref. | | | |
| | White | 0.23 | 0.08 | 8.45 | 1.26 ** [1.08–1.47] | 0.12 | 0.12 | 0.94 | 1.13 [0.88–1.44] |
| | Blue | 0.38 | 0.06 | 38.82 | 1.46 *** [1.3–1.65] | −0.22 | 0.10 | 4.43 | 0.80 * [0.65–0.99] |
| | Pink | 0.15 | 0.09 | 2.73 | 1.17 [0.97–1.4] | 0.01 | 0.15 | 0.00 | 1.01 [0.75–1.36] |
| **Income Quintile (Individual)** | Lower | Ref. | | | | Ref. | | | |
| | Lower-middle | 0.12 | 0.08 | 2.40 | 1.13 [0.97–1.31] | −0.13 | 0.14 | 0.84 | 0.88 [0.67–1.16] |
| | Middle | 0.00 | 0.08 | 0.00 | 1.00 [0.86–1.17] | −0.10 | 0.14 | 0.54 | 0.90 [0.69–1.18] |
| | Upper-middle | −0.83 | 0.08 | 1.10 | 0.92 [0.79–1.07] | −0.02 | 0.13 | 0.02 | 0.98 [0.75–1.28] |
| | Upper | 0.01 | 0.08 | 0.01 | 1.01 [0.86–1.18] | −0.06 | 0.13 | 0.21 | 0.94 [0.71–1.23] |
| **Drinking Status** | Abstained in the last year | Ref. | | | | Ref. | | | |
| | Once a month | −0.06 | 0.07 | 0.64 | 0.94 [0.82–1.09] | −0.05 | 0.12 | 0.16 | 0.95 [0.75–1.21] |
| | 2–4 times a month | 0.10 | 0.07 | 1.81 | 1.11 [0.96–1.28] | 0.12 | 0.13 | 0.89 | 1.13 [0.88–1.45] |
| | At least twice a week | 0.20 | 0.08 | 5.82 | 1.21 * [1.04–1.42] | 0.15 | 0.15 | 1.05 | 1.16 [0.87–1.54] |
| **Smoking Status** | Never smoked | Ref. | | | | Ref. | | | |
| | Ex-smoker | 0.77 | 0.06 | 152.96 | 2.16 *** [1.91–2.44] | −0.29 | 0.13 | 5.07 | 0.75 * [0.58–0.96] |
| | Current smoker | 0.73 | 0.07 | 117.83 | 2.07 *** [1.82–2.36] | −0.45 | 0.15 | 9.36 | 0.63 ** [0.47–0.85] |
| **Perceived Stress Level** | Almost Never | Ref. | | | | Ref. | | | |
| | Occasionally | −0.18 | 0.08 | 5.06 | 0.84 * [0.71–0.98] | −0.03 | 0.15 | 0.05 | 0.97 [0.72–1.3] |
| | Often | −0.40 | 0.09 | 20.37 | 0.67 *** [0.56–0.8] | 0.25 | 0.16 | 2.40 | 1.28 [0.94–1.74] |
| | Very Often | −0.70 | 0.13 | 28.52 | 0.5 *** [0.38–0.64] | 0.66 | 0.20 | 10.43 | 1.93 ** [1.29–2.87] |
| **Fasting Glucose (md/dL)** | Normal | Ref. | | | | Ref. | | | |
| | Borderline | 0.16 | 0.06 | 7.17 | 1.17 ** [1.04–1.31] | −0.44 | 0.13 | 12.19 | 0.64 *** [0.5–0.82] |
| | Diabetes | 0.05 | 0.15 | 0.12 | 1.05 [0.79–1.41] | −0.52 | 0.42 | 1.57 | 0.59 [0.26–1.34] |
| **HbA1c (%)** | Normal | Ref. | | | | Ref. | | | |
| | Borderline | 0.14 | 0.05 | 6.71 | 1.15 * [1.04–1.28] | −0.17 | 0.09 | 3.4 | 0.84 [0.7–1.01] |
| | Diabetes | −0.04 | 0.14 | 0.07 | 0.96 [0.73–1.28] | −0.43 | 0.39 | 1.19 | 0.65 [0.3–1.41] |
| **Total Cholesterol (md/dL)** | Normal | Ref. | | | | Ref. | | | |
| | Borderline | 0.05 | 0.05 | 0.93 | 1.05 [0.95–1.17] | −0.01 | 0.10 | 0.00 | 0.99 [0.82–1.2] |
| | Hyperlipidemia | 0.03 | 0.08 | 0.14 | 1.03 [0.88–1.21] | −0.11 | 0.15 | 0.51 | 0.9 [0.67–1.21] |
| **HDL Cholesterol (md/dL)** | Normal | Ref. | | | | Ref. | | | |
| | Low | 0.26 | 0.07 | 12.32 | 1.30 *** [1.12–1.5] | −0.71 | 0.22 | 10.94 | 0.49 *** [0.32–0.75] |
| | High | −0.55 | 0.06 | 79.64 | 0.58 *** [0.51–0.65] | 0.36 | 0.09 | 15.86 | 1.44 *** [1.2–1.72] |
| **Hypertension** | Normal | Ref. | | | | Ref. | | | |
| | Prehypertension | 0.4 | 0.05 | 56.17 | 1.5 *** [1.35–1.66] | −0.51 | 0.11 | 21.79 | 0.6 *** [0.48–0.74] |
| | Hypertension | 0.33 | 0.08 | 18.16 | 1.4 *** [1.2–1.63] | −0.53 | 0.18 | 8.21 | 0.59 ** [0.41–0.85] |
| **Triglyceride (md/dL)** | Normal | Ref. | | | | Ref. | | | |
| | Borderline | 0.42 | 0.07 | 32.82 | 1.52 *** [1.32–1.76] | 0.18 | 0.15 | 1.35 | 1.19 [0.89–1.61] |
| | Danger | 0.28 | 0.08 | 13.49 | 1.32 *** [1.14–1.53] | −0.09 | 0.19 | 0.25 | 0.91 [0.63–1.31] |
| **MVPA** | Met | Ref. | | | | Ref. | | | |
| | Not met | −0.26 | 0.06 | 22.77 | 0.77 *** [0.69–0.86] | 0.26 | 0.10 | 6.53 | 1.3 * [1.06–1.58] |

*(Continued)*

**Table 3.** (Continued)

| Variables | | Group B | | | | Group C | | | |
|---|---|---|---|---|---|---|---|---|---|
| | | B | S.E | Wald | O.R (95% CI) | B | S.E | Wald | O.R (95% CI) |
| **Sleep Time (hours/day)** | 7 - 9 hours | Ref. | | | | Ref. | | | |
| | Under 7 hours | 0.11 | 0.05 | 4.93 | 1.12 * [1.01–1.23] | −0.02 | 0.09 | 0.06 | 0.98 [0.82–1.17] |
| | Over 7 hours | −0.1 | 0.11 | 0.83 | 0.9 [0.72–1.13] | 0.30 | 0.16 | 3.45 | 1.35 [0.98–1.85] |
| **Sedentary Behavior** | Low | Ref. | | | | Ref. | | | |
| | Middle | 0.08 | 0.07 | 1.57 | 1.09 [0.95–1.24] | 0.21 | 0.12 | 3.20 | 1.24 [0.98–1.16] |
| | High | −0.11 | 0.06 | 3.22 | 0.90 [0.80–1.01] | 0.13 | 0.11 | 1.41 | 1.13 [0.92–1.39] |

**Note.** *** $p < 0.001$, ** $p < 0.01$, * $p < 0.05$, SE: Standard Error, OR: Odd Ratio, CI: Confidence Interval, PA: Physical Activity, SB: Sedentary Behavior, MVPA: Moderate-to-Vigorous Physical Activity.

Multinomial logistic regression was performed using Group A as the reference category.

individuals who perceived themselves as heavier were less likely to belong to the hyperlipidemia or hypertension groups. These results suggest that the effect of SBIP on metabolic health is more nuanced than merely considering PA or SB alone. Specifically, individuals who underestimate their body image may be more susceptible to negative metabolic conditions, while those who overestimate their body image may engage in healthier behaviors. This underscores the importance of integrating SBIP into intervention strategies aimed at promoting PA and addressing metabolic health issues. Furthermore, managing the psychological factors associated with body image perception could effectively improve metabolic health.

Interestingly, individuals who perceived themselves as heavier than their actual BMI were less likely to meet the WHO's PA recommendations, exhibiting the lowest average levels of MVPA among the groups studied. These findings align with prior research indicating that negative body image perceptions correlate with lower PA levels and reduced sports participation. Notably, the group with lower SBIP relative to their actual BMI had a higher average

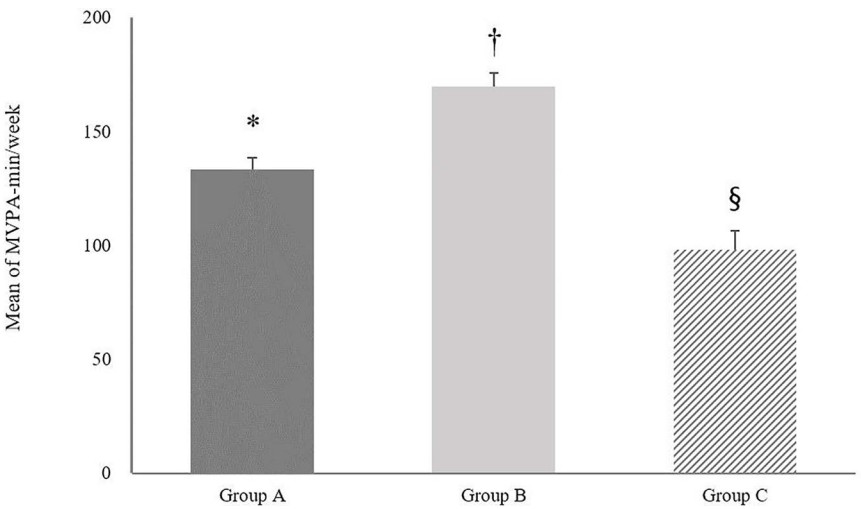

**Fig 2. Average moderate-to-vigorous physical activity minutes per week for three groups.** Group A BMI = SBIP, Group B: BMI > SBIP, Group C: BMI < SBIP. Error bars represent the standard error of the mean. * indicates a statistically significant difference between group A and group C. † indicates a statistically significant difference between group A and group B. § indicates a statistically significant difference between group B and group C. Statistical significance was determined at the 0.05 level.

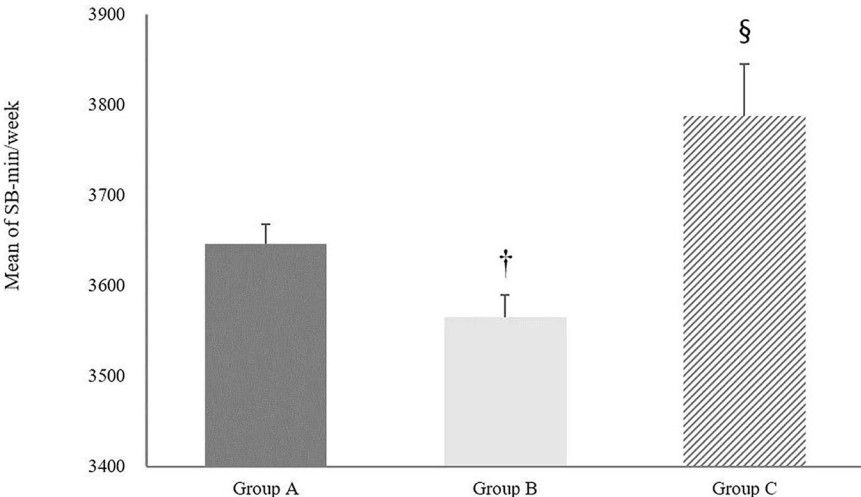

**Fig 3. Average sedentary behavior minutes per week for three groups.** Group A BMI = SBIP, Group B: BMI > SBIP, Group C: BMI < SBIP. Error bars represent the standard error of the mean. † indicates a statistically significant difference between group A and group B. § indicates a statistically significant difference between group B and group C. Statistical significance was determined at the 0.05 level.

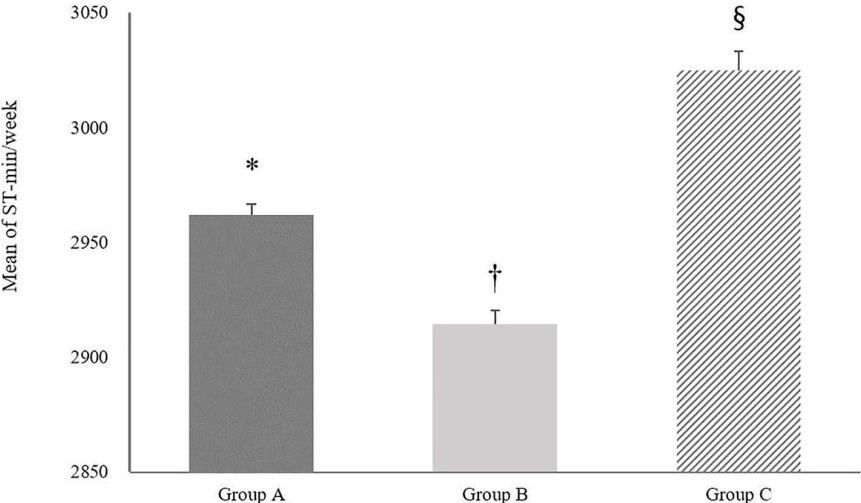

**Fig 4. Average sleep duration minutes per week for three groups.** Group A BMI = SBIP, Group B: BMI > SBIP, Group C: BMI < SBIP. Error bars represent the standard error of the mean. * indicates a statistically significant difference between group A and group C. † indicates a statistically significant difference between group A and group C. § indicates a statistically significant difference between group B and group C. Statistical significance was determined at the 0.05 level.

BMI, while individuals with higher BMIs demonstrated greater SB and ST compared to those with lower BMIs [40,41]. This contrasts with existing literature that generally associates higher BMI with lower PA and increased SB and ST. This discrepancy implies that SBIP may significantly influence health behaviors. Furthermore, a study by Tebar suggested that extended sedentary time is linked to higher body dissatisfaction, indicating that a sedentary lifestyle may adversely affect body image [33]. Overall, our study reveals that individuals who perceive themselves as heavier than

their actual BMI are more likely to engage in lower levels of PA, fall short of meeting WHO PA recommendations, and exhibit greater SB. This suggests that SBIP plays a critical role in influencing health behaviors, potentially exerting a greater influence than BMI alone.

Additionally, we found that individuals who perceived their SBIP as higher than their actual BMI were more likely to have attained higher levels of education. This relationship between educational attainment and SBIP has been underexplored, and our results provide novel evidence that educational levels may play a critical role in shaping SBIP. Potential mechanisms include greater health literacy, increased exposure to health-related information, or heightened awareness of societal body image standards among more educated individuals, which may lead to more critical self-evaluation of body image. Moreover, participants with a higher SBIP than their actual BMI reported more frequent experiences of stress, aligning with Warren's findings that women experiencing body dissatisfaction are more prone to elevated stress levels [42]. This link could be driven by various factors, including concerns about obesity, societal pressures to conform to ideal body standards, and challenges related to weight control. The persistent stress associated with body dissatisfaction may exacerbate other health issues, including metabolic health challenges and mental health disorders, underscoring the importance of addressing SBIP in holistic health interventions. Numerous studies have demonstrated a robust correlation between SBIP and stress, suggesting that the psychological burden of body image concerns significantly impacts mental health. These findings emphasize the need for targeted mental health care strategies addressing SBIP, particularly in populations with higher education levels who may be more susceptible to stress due to heightened body image awareness. Interventions aimed at improving body satisfaction and managing stress could enhance overall health outcomes, mitigating the negative impacts of SBIP discrepancies on both physical and mental health.

Our study also found that individuals with a higher SBIP than their actual BMI were less likely to engage in PA, consistent with Hwang's findings that positive body shape perception correlates with higher PA levels [43]. Conversely, participants who perceived themselves as thinner than their actual BMI exhibited higher PA levels, reduced SB, and insufficient ST. This aligns with evidence suggesting that greater PA and lower SB positively affect stress reduction. Furthermore, NBIP can diminish exercise motivation and PA, indicating that lower SBIP compared to BMI may hinder health behavior changes. Additionally, participants perceiving themselves as overweight reported increased ST, as supported by Akram's research linking higher SBIP with adverse sleep patterns [44]. Kosa's findings reinforce that both excessive and insufficient sleep can adversely affect health, highlighting the importance of fostering a positive SBIP for better sleep and overall health [45]. Overall, this study emphasizes the significant impact of both SBIP and BMI on PA and health outcomes. Thus, integrating both factors into PA promotion and stress reduction programs is beneficial. Enhancing positive body shape awareness through targeted interventions may lead to more effective health promotion strategies and improved overall health.

This study has several strengths and limitations. The main strength is its identification of the relationship between various health-related factors according to the degree of difference between BMI and SBIP. While previous studies have analyzed BMI and SBIP independently, this research provides a more complex understanding of how their differences influence PA and related outcomes. However, the study's limitations include its focus on Korean subjects, which may limit the generalizability of results to diverse populations. Additionally, while the data were collected from a reliable national organization, reliance on self-reported questionnaires may introduce inaccuracies due to respondents' subjective judgment or memory. Furthermore, the cross-sectional design of this study presents a limitation in establishing a causal relationship between the discrepancy of BMI and SBIP and various health indicators. As a cross-sectional study analyzes data at a single point in time, it is challenging to determine temporal precedence or causality between variables. Therefore, future longitudinal studies are needed to address this limitation and provide clearer insights into the causal relationships among these factors. Furthermore, future research should aim to enhance reliability through real-time PA measurement using devices such as wearables, objective data collection based on biometric information, and linkage with medical records to supplement self-report questionnaire limitations.

 

Additionally, conducting multinational studies involving various races, cultures, and environments could enhance the external validity of the findings.

## 5. Conclusion

In conclusion, this study reveals that discrepancies between actual BMI and SBIP significantly impact various health indicators, including metabolic health indicators, PA, SB, ST, and stress levels. The findings suggest that SBIP may have a more profound influence on health behaviors than BMI alone, particularly in shaping PA and SB patterns. Individuals who perceive themselves as heavier than their actual BMI tend to engage in lower levels of PA, exhibit greater SB, and report higher stress levels. These results underscore the importance of incorporating SBIP into health promotion strategies aimed at enhancing PA, addressing metabolic concerns, and managing stress. Addressing SBIP through targeted interventions could lead to more effective health outcomes. Future research should utilize objective measures and consider diverse populations to improve the generalizability and accuracy of these findings.

## Acknowledgments

We want to thank for participants who took part in our experiments.

## Author contributions

**Conceptualization:** Ho-Jun Kim, Jung-Min Lee.

**Data curation:** Ho-Jun Kim.

**Formal analysis:** Ho-Jun Kim, In-Whi Hwang, Kyu-Ri Hong, Hae-Young Chung, Jung-Min Lee.

**Funding acquisition:** Jung-Min Lee.

**Investigation:** Ho-Jun Kim, In-Whi Hwang, Kyu-Ri Hong, Hae-Young Chung, Jung-Min Lee.

**Methodology:** In-Whi Hwang, Kyu-Ri Hong, Jung-Min Lee.

**Supervision:** Jung-Min Lee.

**Validation:** In-Whi Hwang, Kyu-Ri Hong, Hae-Young Chung, Jung-Min Lee.

**Writing – original draft:** Ho-Jun Kim, Jung-Min Lee.

**Writing – review & editing:** Ho-Jun Kim, In-Whi Hwang, Kyu-Ri Hong, Hae-Young Chung, Jung-Min Lee.

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
