## [Decision Letter · Decision Letter 0]

25 Mar 2025

Dear Dr. Lee,

Thank you for submitting your manuscript to PLOS ONE. After careful consideration, we feel that it has merit but does not fully meet PLOS ONE’s publication criteria as it currently stands. Therefore, we invite you to submit a revised version of the manuscript that addresses the points raised during the review process.

**ACADEMIC EDITOR:**

**Dear Authors:**

**Overall,** the research is well-structured, with clear objectives and a strong methodological framework. However, some areas need further clarification and refinement. Addressing these points will enhance the manuscript’s clarity, rigor, and impact.

The calculation of moderate-to-vigorous physical activity (MVPA) relies on self-reported data. Were any objective measures (e.g., accelerometer data) considered? If not, discuss the potential biases introduced by self-reported activity levels.The study uses self-reported stress levels, but additional details on the validity and reliability of this measure would strengthen the findings. If validated scales were used, cite them appropriately.The role of SES is discussed, but the analysis could be expanded. Were any sensitivity analyses conducted to control for potential SES confounders? If not, this should be acknowledged as a limitation.In the multinomial logistic regression analysis, please clarify whether adjustments were made for potential confounders such as age and gender.The manuscript is well-written, but minor grammatical and typographical errors should be addressed to improve readability.

We look forward to receiving your revised manuscript.

Kind regards,

Ozra Tabatabaei-Malazy

Academic Editor

PLOS ONE

Journal Requirements:

This work was supported by the Ministry of Education of the Republic of Korea and the

National Research Foundation of Korea (NRF-2021S1A5A8064404)

4. Please remove all personal information, ensure that the data shared are in accordance with participant consent, and re-upload a fully anonymized data set.

Reviewers' comments:

Reviewer's Responses to Questions

**Comments to the Author**

1. Is the manuscript technically sound, and do the data support the conclusions?

Reviewer #1: Yes

Reviewer #2: Yes

2. Has the statistical analysis been performed appropriately and rigorously?

Reviewer #1: No

Reviewer #2: Yes

3. Have the authors made all data underlying the findings in their manuscript fully available?

Reviewer #1: Yes

Reviewer #2: Yes

4. Is the manuscript presented in an intelligible fashion and written in standard English?

Reviewer #1: Yes

Reviewer #2: Yes

Reviewer #1: This article highlights the relationship between objective and subjective body composition, or the body fitness concept, which is a major concern and an important component of lifestyle medicine and mental and physical health. Synthesizing the extensive population-base data can provide valuable insights into these issues. I have made some comments to improve this work.

1. Do not use the term “risk” in cross-sectional studies, as we cannot draw causal conclusions in this design due to the unclear temporal precedence. Please review all manuscripts accordingly.

2. This is too long an introduction. Please shorten it into two or three paragraphs related to the main concepts, their relationship or other resulting health issues.

3. You had excluded the non-answers and answered them with “I don’t know”. Please provide more detail on these groups, such as missing tables or charts, to account for any bias or missing patterns in your work

4. What is the reason for excluding underweight people? Did you perform any sensitivity analysis with this group to ensure the validity of your results?

5. Please mention the cut-off point for obesity or overweight in the "Measurement" section in the method; the reference you cite in line 161 does not match the values given in line 185

6. Please clarify more about this definition, “Occupational status (white-collar, blue-collar, pink-collar, not classified),”

7. Lines 189 and 190 are duplicated in the section on method. You have already mentioned this in section 2.1

8. The explanation of the weighting method in the calculation of total activity was misleading, please simplify.

9. In the statistical method, you should explain the independent and dependent variables in the multinomial logistic regression to provide more clarity.

10. According to your results in Table 2, you have little data in these categories, especially in group c, so the chi-square test is not valid and appropriate to assess the transparency between groups in this case.

11. The results reported in Table 3 are very vague as there is no clear methodology for variable selection, adjustments, co-linearity checking and interpretation of the multinomial models. Please revise the results and provide the necessary explanations for more clarity.

Reviewer #2: -The title of the article does not reflect its content and aim

Abstract:

- Abstract should be structured based on journal format

- Numerical values needs to be added to the results

- All abbreviations needs to be defined (e.g OR, HDL, etc.)

Introduction:

- This sections is too long. unnecessary information needs to be removed or summarized.

- line 67: define abbreviations

- line 79: mention the exact number for proportion instead the word "significant"

- What is the gap of knowledge?

Method:

- One of the problems of the method section is that first there is a summary of methodology and then a comprehensive explanation of it. I suggest merging them and do not repeating them. For instance, the authors have explained about ABC categories and then in another section gave a complete explanation about categorization.

- Add proper reference for the first paragraph of 2.1.

- Why didn't the authors use previous cycles of KNHANES data?

- More details regarding physical examinations and data collections and measurements should be added ( height, weight, etc.

- Mention the inclusion and exclusion criteria and add a flow diagram for it.

- line 141: SP < BMI? are you sure?

- what do you mean by white, blue, and pink collar?

- what is the definition of ex-smoker?

- why didn't the author use alcohol volume rather than frequency of drinking?

Results:

- lines 301-321: there is no need to mention X2 values in the main text.

- Table 2: Mention p values rather than X2 values

Discussion: Mention the cross-sectional design of the study as a limitationn for causal relationship

**Do you want your identity to be public for this peer review?** For information about this choice, including consent withdrawal, please see our Privacy Policy

Reviewer #1: No

Reviewer #2: No

---

## [Author Response · Author response to Decision Letter 1]

7 May 2025

We have carefully addressed all reviewer and editor comments. A detailed, point-by-point response is provided in the uploaded "Response to Reviewers" document. All revisions have been clearly marked in the manuscript.

---

## [Decision Letter · Decision Letter 1]

13 May 2025

Dear Dr. Lee,

Thank you for submitting your manuscript to PLOS ONE. After careful consideration, we feel that it has merit but does not fully meet PLOS ONE’s publication criteria as it currently stands. Therefore, we invite you to submit a revised version of the manuscript that addresses the points raised during the review process.

We look forward to receiving your revised manuscript.

Kind regards,

Ozra Tabatabaei-Malazy

Academic Editor

PLOS ONE

Journal Requirements:

Reviewers' comments:

Reviewer's Responses to Questions

**Comments to the Author**

Reviewer #1: (No Response)

Reviewer #2: (No Response)

2. Is the manuscript technically sound, and do the data support the conclusions?

Reviewer #1: Yes

Reviewer #2: (No Response)

3. Has the statistical analysis been performed appropriately and rigorously?

Reviewer #1: Yes

Reviewer #2: (No Response)

4. Have the authors made all data underlying the findings in their manuscript fully available?

Reviewer #1: Yes

Reviewer #2: (No Response)

5. Is the manuscript presented in an intelligible fashion and written in standard English?

Reviewer #1: Yes

Reviewer #2: (No Response)

Reviewer #1: many thanks for this improved revision of the manuscript and that nearly all comments have been addressed:

as I noted in the previous revisions, the introduction was too long and should be shortened and summarized to make it readable for other researchers.

If you report the results of multinomial logistic regression in the form of OR, you must not use the prevalence rate, but use the correct term such as odds or chance of disease.

the exclusion of underweight people is not properly addressed! You simply deleted some sentences in this area of concern during the revision.

Ex-smoking is not yet clearly defined,

Th method of physical activity assessing was not cited as a valid reference. Do you have a question or a valid measure for the method used?

Reviewer #2: (No Response)

**Do you want your identity to be public for this peer review?** For information about this choice, including consent withdrawal, please see our Privacy Policy

Reviewer #1: No

Reviewer #2: No

---

## [Author Response · Author response to Decision Letter 2]

17 Jun 2025

Thank you for the opportunity to revise our manuscript. We have carefully reviewed all comments from the editor and reviewers and have revised the manuscript accordingly. Detailed responses to each comment are provided in the attached Response to Reviewers document. We sincerely appreciate the constructive feedback, which helped us improve the quality and clarity of our work.

---

## [Decision Letter · Decision Letter 2]

30 Jun 2025

Exploring the Interplay Between BMI, Subjective Body Image Perception, and Health Behaviors: A cross-sectional study

PONE-D-25-05753R2

Dear Dr. Lee,

We’re pleased to inform you that your manuscript has been judged scientifically suitable for publication and will be formally accepted for publication once it meets all outstanding technical requirements.

Kind regards,

Ozra Tabatabaei-Malazy

Academic Editor

PLOS ONE

Additional Editor Comments (optional):

Reviewers' comments:

Reviewer's Responses to Questions

**Comments to the Author**

Reviewer #1: All comments have been addressed

2. Is the manuscript technically sound, and do the data support the conclusions?

Reviewer #1: Yes

3. Has the statistical analysis been performed appropriately and rigorously?

Reviewer #1: Yes

4. Have the authors made all data underlying the findings in their manuscript fully available?

Reviewer #1: Yes

5. Is the manuscript presented in an intelligible fashion and written in standard English?

Reviewer #1: Yes

Reviewer #1: Thank you for the revised version and for addressing all the required comments. And know ready to publication .

**Do you want your identity to be public for this peer review?** For information about this choice, including consent withdrawal, please see our Privacy Policy

Reviewer #1: No

---

## [Editor Report · Acceptance letter]

PONE-D-25-05753R2

PLOS ONE

Dear Dr. Lee,

I'm pleased to inform you that your manuscript has been deemed suitable for publication in PLOS ONE. Congratulations! Your manuscript is now being handed over to our production team.

Kind regards,

on behalf of

Dr. Ozra Tabatabaei-Malazy

Academic Editor

PLOS ONE